Ecological associations of the coastal marsh periwinkle snail Littoraria irrorata: field and laboratory evidence of vegetation habitat preferences

Klinges David H. 1 2
Martin Charles W. 3 4
Roberts Brian J. broberts@lumcon.edu 5
1 Department of Biological Sciences, Dartmouth College , Hanover , NH , United States of America
2 School of Natural Resources and Environment, University of Florida , Gainesville , FL , United States of America
3 Stokes School of Marine & Environmental Sciences, University of South Alabama , Mobile , AL , United States of America
4 Dauphin Island Sea Lab , Dauphin Island , AL , United States of America
5 Louisiana Universities Marine Consortium , Chauvin , LA , United States of America
Brygadyrenko Viktor
Electronic publication date: 2025 Mar 12
Publication date: 2025
Volume: 13
Electronic Location ID: e19071
Received 2025 Jan 9; Accepted 2025 Feb 10
Copyright: ©2025 Klinges et al.
Copyright year: 2025
Copyright holder: Klinges et al.
License: This is an open access article distributed under the terms of the Creative Commons Attribution License, which permits unrestricted use, distribution, reproduction and adaptation in any medium and for any purpose provided that it is properly attributed. For attribution, the original author(s), title, publication source (PeerJ) and either DOI or URL of the article must be cited.
License URL: https://creativecommons.org/licenses/by/4.0/

Keywords: Salt marsh, Gulf Coast, Spartina alterniflora, Distichlis spicata, Juncus roemerianus, Spartina patens, Experiment, Snail density

Funding: The Gulf of Mexico Research Initiative to the Coastal Waters Consortium The NSF REU Site Program OCE-1757887 The National Science Foundation Graduate Research Fellowship DGE-1842473 This research was made possible by a grant from the Gulf of Mexico Research Initiative to the Coastal Waters Consortium. The involvement of David Klinges was made possible by a grant from the NSF REU Site Program (OCE-1757887) to Brian Roberts and LUMCON, and David Klinges was further supported by the National Science Foundation Graduate Research Fellowship (DGE-1842473) during the writing of this manuscript. The funders had no role in the design, execution, or analyses of this project. The funders had no role in study design, data collection and analysis, decision to publish, or preparation of the manuscript.

==============================
Coastal salt marshes serve as the margin between terrestrial and marine biomes, provide a variety of important services, and are dynamic ecosystems characterized by keystone species that shape trophic networks. In coastal salt marshes of the Atlantic and Gulf Coasts of the United States, marsh periwinkle snails (Littoraria irrorata) exhibit high abundance and form critical trophic pathways as important herbivores and detritivores. Specifically, snails forage on Spartina alterniflora and associated fungal growth, for which L. irrorata may act as a top-down control on plant growth. Yet, L. irrorata occupies other salt marsh plants, suggesting its habitat niche may be broader than previously reported. Here, we documented snail densities and size distributions in a Louisiana (USA) salt marsh composed of multiple marsh graminoids and report the results of behavioral choice experiments designed to test snail habitat preferences as a potential mechanism underlying their field distribution. We observed higher snail densities on S. alterniflora stalks (283 snails m−2) than other plant species, however, snails were highly abundant on S. patens (116 snails m−2), Juncus roemerianus (95 snails m−2), and Distichlis spicata (57 snails m−2) with densities comparable or higher on all species than reported on S. alterniflora in other studies along the U.S. Atlantic and Gulf coasts. Snails found on S. alterniflora and J. roemerianus, both plants with tall and rigid stalks, were also larger than snails found on other plant species. In species preference experiments, snails preferred S. alterniflora over S. patens and D. spicata, but no clear preferences were observed between S. alterniflora and J. roemerianus, nor between any combinations of S. patens, D. spicata, and J. roemerianus. Finally, we found that snails preferred senescing and dead S. alterniflora tissue over fresh S. alterniflora. Interpreting these results in tandem, this study suggests L. irrorata snails have consistent patterns of field distributions that match their habitat preferences, and future studies should test potential processes driving snail habitat selection, such as dietary habits and predator refugia (i.e., climbing sturdy stalks to avoid aquatic predators). Considering the abundance and trophic role of L. irrorata in coastal salt marshes, snail behavior may be a key modulator for salt marsh trophic networks.

Introduction

Coastal salt marshes provide a variety of ecosystem services, including serving as a basal food source for a productive food web (McCann et al., 2017), providing structural refuge for juveniles of many commercially and recreationally important species (Peterson & Turner, 1994; Able et al., 2015), protecting inland areas from high intensity storms (Costanza et al., 2008), and decreasing eutrophication via nutrient cycling and removal (Hopkinson & Giblin, 2008). Marsh ecosystems can be stressful, owing to dynamic environmental conditions (e.g., temperature, salinity, inundation, dissolved oxygen, etc.) that can change rapidly. As a result, species assemblages include tolerant flora and fauna capable of withstanding extreme conditions. Marsh plants exhibit numerous adaptations for survival in these areas, including salt-excreting glands, resistance to flooding conditions, and broad thermal tolerances. Plant diversity within salt marshes can also be low, and this highlights the need for better understanding of the roles each plant species plays in supporting food webs. For example, omnivorous snails comprise highly abundant biomass pools and important trophic intermediates, facilitating the transfer of energy from basal plant production to higher trophic levels (Hamilton, 1976; Silliman & Zieman, 2001; McCann et al., 2017).

The salt marsh periwinkle snail (Littoraria irrorata) inhabits, and is often the dominant snail in, salt marshes of the Atlantic and Gulf coasts of the United States. This snail resides in emergent vegetation on the marsh platform where it displays distinct behavior of climbing plant stems at high tide to avoid predation by aquatic predators from below (Warren, 1985; Carroll, Church & Finelii, 2018), as a mechanism of thermoregulation (Williams & Appel, 1989; Henry, McBride & Williams, 1993), and to facilitate fungal invasion on plant leaves for subsequent consumption (Silliman & Newell, 2003). Littoraria irrorata graze primarily on Spartina alterniflora compared to other plant species (e.g., Hendricks, Mossop & Kicklighter, 2011; Sieg et al., 2013), and especially graze senesced rather than live S. alterniflora leaves (Bärlocher & Newell, 1994). Littoraria irrorata populations influence a variety of marsh ecosystem components including vegetation, microbial communities, organic matter and nutrient cycling, and marsh-estuarine food webs (Zengel et al., 2017). While some studies indicate L. irrorata exerts top-down control on plant aboveground biomass and productivity (Silliman & Zieman, 2001), others have found no support for top-down control of marsh plant productivity (Kiehn & Morris, 2009). Ecologists have most frequently associated study of L. irrorata with S. alterniflora that commonly define coastal salt marshes (e.g., Hamilton, 1976; Silliman & Zieman, 2001; Zengel et al., 2017; Rietl, Sorrentino & Roberts, 2018), and therefore the trophic role of L. irrorata has typically been considered linked (and largely limited) to the abundance and distribution of S. alterniflora (Silliman & Zieman, 2001; McFarlin et al., 2015).

Coastal salt marshes are heterogeneous ecosystems containing mosaics of plant species arranged in patches that reflect variation in local conditions (e.g., elevation/inundation frequency, salinity, soil properties, etc.) and interspecific competition (Pennings, Grant & Bertness, 2005). In addition to the dominant S. alterniflora, coastal marshes along the United States Gulf Coast also contain patches of the macrophytes S. patens, Juncus roemerianus, and Distichlis spicata. Salt marsh plants in this region exhibit different responses to hydrologic alterations including flooding frequency and salinity stress (Jones et al., 2016). As a result, future changes in climate, inundation, and salinity regimes are predicted to change plant community structure in coastal salt marshes. Given this, developing a more comprehensive understanding of plant-animal interactions involving abundant marsh plants is critical information for predicting future food webs and trophic structure.

High densities of L. irrorata snails have been reported in salt marshes along the Gulf and Atlantic coasts of the United States (Rietl, Sorrentino & Roberts, 2018). However, few studies have quantified the densities or lengths of snails on different types of salt marsh vegetation (but see Hughes, 2012; Faillon, Wittyngham & Johnson, 2020). Further, little information is available on snail preferences for different vegetation types that vary in relative abundances across the marsh landscape. Here, we used empirical and experimental investigations to determine the abundance and habitat preferences of L. irrorata snails across the marsh landscape. We provide field-based estimates of snail density and size distributions across multiple plant species in a Louisiana salt marsh. In addition, we performed controlled laboratory experiments to test snail preferences for: (1) various species of marsh plants, (2) stages of S. alterniflora senescence, and (3) plant tissue against structural controls. We predicted snail field distributions would be reflected in experimental choice tests and that snails would prefer senesced over live plants and live plants over wooden structural controls. The overarching goal of this research was to gain insight into snail distributions and preference patterns to develop a better understanding of the marsh ecosystem and food web. Portions of this text were previously published as part of a preprint (Klinges, Martin & Roberts, 2024).

Materials & Methods

Snail densities and size distributions

We quantified snail densities and lengths at a well-studied marsh site (Able et al., 2015; Marton et al., 2015; Bernhard et al., 2019; Rietl, Sorrentino & Roberts, 2018; Keppeler et al., 2021) along the northwestern shore of Bay Batiste (29.4759° N, 89.8543° W) on seven (snail length) to nine (snail density) dates between May 2016 and January 2018 (Table 1). On each date, we collected snails in at least three replicate 0.25 × 0.25 m quadrats from within each of four monoculture vegetation types (S. alterniflora, S. patens, D. spicata, and J. roemerianus). All snails were kept on ice during transport, and stored at 4 °C to await processing, which was completed within 48 h of collection. In the laboratory, we rinsed, cleaned, and counted all snails before measuring shell length (mm) using digital calipers, which had accuracy and precision of 0.01 mm.

Table 1 Littoraria irrorata field densities across time.

Mean and standard error (SE) of Littoraria irrorata densities (snails m−2) in 0.25 m × 0.25 m single-species plots (n = 3 − 5) of Spartina alterniflora, Spartina patens, Juncus roemerianus, or Distichlis spicata on each of nine sampling dates in a salt marsh in Bay Batiste, LA. Bold values represent the overall mean and standard error of snail densities across all sampling dates for each plant species.

Date	S. alterniflora	S. patens	J. roemerianus	D. spicata	
	Mean	SE	Mean	SE	Mean	SE	Mean	SE	
5/16/2016	345.6	68.0	96.0	66.6	112.0	24.4	64.0	16.0	
6/15/2016	316.0	65.8	272.0	84.7	288.0	9.2	117.3	32.4	
8/16/2016	293.3	129.4	117.3	64.9	106.7	5.3	32.0	9.2	
10/11/2016	320.0	92.9	90.7	50.9	26.7	14.1	48.0	16.0	
2/1/2017	406.4	147.7	64.0	16.0	32.0	24.4	58.7	29.7	
5/9/2017	176.0	78.9	202.7	101.3	85.3	19.2	53.3	14.1	
9/7/2017	128.0	42.3	85.3	10.7	53.3	23.2	80.0	32.0	
10/26/2017	298.7	78.6	10.7	10.7	96.0	40.2	48.0	24.4	
1/25/2018	309.3	101.3	101.3	29.7	58.7	10.7	10.7	10.7	
Mean	288.1	28.4	115.6	25.8	95.4	26.2	56.9	10.0	

Habitat preference experiments

We experimentally determined snail habitat preferences following methods established in previous habitat choice studies (Martin, 2017; Martin et al., 2020). All trials were performed in 20-liter (15 cm × 30 cm × 20 cm) arenas, each containing 100 mL of 9 psu seawater (Instant Ocean®, Instant Ocean Spectrum Brands), a typical salinity for Gulf of Mexico salt marshes. We obtained snails and plants from marshes near the Louisiana Universities Marine Consortium (LUMCON)’s DeFelice Marine Center in Cocodrie, Louisiana (USA) (29.2580° N, 90.6629° W), a representative coastline with extensive salt marsh (e.g., Hill & Roberts, 2017). Snails were collected from stands of the four studied species of vegetation and stored in the same cooler for transport so that the source vegetation was randomized. We performed three experiments to test snail preference patterns for: (1) marsh plant species, (2) S. alterniflora state of senescence, and (3) plant matter as opposed to structural controls. In experiment 1, we tested snail preference for plant species by offering a choice between each of the following combination of plants (n = 10 for each combination): S. alterniflora vs. S. patens, S. alterniflora vs. D. spicata, S. alterniflora vs. J. roemerianus, S. patens vs. D. spicata, S. patens vs. J. roemerianus, and D. spicata vs. J. roemerianus. In experiment 2, we tested snail preference for different S. alterniflora states of senescence, using all combinations of green (live), yellow (partially senesced), and brown (dead) S. alterniflora, as described by (Graça, Newell & Kneib, 2000) (n = 10 for each combination). Finally, experiment 3 was conducted to determine whether snails preferred any of the four plant species (S. alterniflora, S. patens, D. spicata, and J. roemerianus) to a structural control, which offers a rigid structure to climb and escape aquatic predators, but no viable food source (n = 5 for each single species versus structural control combination).

We cut all plants used in experiments to 15-cm segments (the height of the arena) using only tissue from between the first leaf and final leaf of a stem for S. alterniflora, S. patens, and D. spicata samples, and using tissue at least 10 cm above the exposed base of a blade and at least 10 cm below the tip of a blade for J. roemerianus. We rinsed and mounted plant stalks in polystyrene foam inserts and placed them at each end of the arena. All stalks were standardized to contain equal volume (90 cm3) of each species used in trials (4–15 stalks used per trial). In experiment 3, we used six dowels of approximately equal diameter (one cm) and height (15 cm) as paired plant stalks to offer structural refuge but no viable food source paired with one of the four plant species. Dowels were replaced for every trial to avoid the possibility of chemical cues impacting snail behavior.

We randomly selected six snails (20–25 mm shell length) for use in each trial after starving snails for 48 h. This density is within the natural range of snail densities we observed in this study and reported in Rietl, Sorrentino & Roberts (2018). We placed snails in the middle of arenas, and a camera mounted 30 cm above each arena captured a photograph of the arena interior at five-minute intervals for 12 h. Due to this short trial duration, controls to correct for autogenic or allogenic changes to plant tissues were not necessary, as the amount of decomposition of plant tissue in 12 h was minimal (Roa, 1992). We covered arenas with clear plastic wrap to prevent snail escape while maintaining visibility from above for time-lapse photography. Each trial included six hours of simulated daylight (four white fluorescent lights at 25 °C) and six hours of simulated night (four low-wattage violet fluorescent lights at 20 °C) to account for diurnal differences in snail behavior (Graça, Newell & Kneib, 2000; Iacarella & Helmuth, 2011). To capture snail behavior during night conditions, we marked snail shells with odorless neon fluorescent paint prior to placement in arenas (Fig. S1). We began half of trials as simulated day, and the other half as simulated night, and the order did not affect snail habitat preference (Kruskal-Wallis H2,356 = 178.4, P = 0.295).

We processed snail preferences using time-lapse photography, which we recorded at five-minute intervals to quantify preference patterns (Video S1). We took a conservative approach when determining habitat choice and considered snail preference whenever snails were within the habitat canopy which was defined as within 0.5 cm of habitat. This included not only the plant or dowel but any arena wall adjacent to that habitat (including the arena floor). We calculated the amount of time each of the six snails per trial exhibited a preference for one available habitat versus the opposite available habitat, and then averaged across all time points for a trial (144 time points per trial) to derive the proportion of time exhibiting preference. For example, if six snails on average spent 40% of a trial within the S. alterniflora habitat, and 5% of a trial within the D. spicata habitat, this would suggest a preference for S. alterniflora.

Statistical analyses

In the field survey, a preliminary analysis indicated that snail density and length varied little across sampling dates (one-way ANOVA: p >0.05). As a result, we pooled dates and conducted analyses using only plant species as the predictor variable. Time-pooled snail densities and lengths were both normally distributed. We conducted separate one-way analysis of variance (ANOVA) for response variables of snail density (snails m−2) and length (mm).

For each choice comparison in experiments, we evaluated differences in preference between habitats by conducting a matched pairs t-test of the difference in the proportion time spent within each provided habitat. We arcsine transformed all proportion data derived from habitat preference experiments in order to stabilize variance and reduce the dependency of variance upon the mean, to uphold assumptions of normality for paired t-tests (Sokal & Rolf, 1995). We conducted all statistical analyses in R 3.5.0 (R Core Team, 2018) and with use of the tidyverse package (Wickham et al., 2019) and considered all results significant at p <0.05.

Results

Snail densities and size distributions

We found L. irrorata in plots within each of the four salt marsh vegetation types (S. alterniflora, S. patens, J. roemerianus, and D. spicata) in Bay Batiste, Louisiana on all nine sampling dates between May 2016 and January 2018. Snail density did not differ with time within any of the vegetation types (Table 1). Across all sampling dates, snail density was significantly (p < 0.05) higher on S. alterniflora (mean ± SE = 288.1 ± 28.4 snails m−2) than the other three vegetation types (Fig. 1A), and densities on S. patens (115.6 ± 25.8 snails m−2) and J. roemerianus (95.4 ± 26.2 snails m−2) were 2.5–3.0 times lower than S. alterniflora and 1.7–2.0 times higher than densities on D. spicata (56.9 ± 10.0 snails m−2). Individual snail shell lengths were significantly larger (p < 0.05) for snails collected in J. roemerianus (22.02 ± 0.14 mm) and S. alterniflora (21.84 ± 0.08 mm) than D. spicata (20.99 ± 0.22 mm) and S. patens (20.98 ± 0.14 mm) (Fig. 1B).

Figure 1 Density and size of periwinkle snails on marsh plant species.

(1) Density of snails in 0.25 m × 0.25 m plots of Spartina alterniflora, Spartina patens, Juncus roemerianus, and Distichlis spicata in a salt marsh in Bay Batiste, LA. Values derived from 3–5 plots per vegetation type on nine dates between May 2016 and January 2018. Boxplots show median, 25th and 75th percentiles (lower and upper hinges, respectively), and whiskers extend to largest and smallest values, unless values are greater or less than 1.5 * IQR (outlying points plotted individually). (2) Length (mm) distributions for snails collected on each of the four vegetation species on the same nine dates. Vertical lines represent the median values for each vegetation type. In both panels, different letters indicate statistically significant (p < 0.05) differences among vegetation types. Snail densities were higher in S. alterniflora stands than in other vegetation, and snails were longer in S. alterniflora and J. roemerianus stands than S. patens and D. spicata stands.

Experiment 1: species preference

Littoraria irrorata snails used in this experiment exhibited clear and significant preferences between marsh plant species (Fig. 2, Table 2). Snails occupied S. alterniflora more often than two of the three other common marsh plant species: on average, we found snails on S. alterniflora 13.6 and 4.3 times more often than on S. patens and D. spicata, respectively. However, there was no significant difference in time spent on S. alterniflora (3.35 ± 6.79%) and time spent on J. roemerianus (3.90 ± 5.56%). We also found snails 12.2 times more often, on average, on D. spicata (2.25 ± 4.01%) than on S. patens (0.183 ± 0.348%; Fig. 2D), but this difference was not significantly different (p = 0.060, Table 2). Snails did not show a significant preference for J. roemerianus compared to either S. patens or D. spicata (Figs. 2E, 2F).

Figure 2 Snail preferences for different marsh plant species.

Percent of time snails spent on (A) S. patens vs. S. alterniflora, (B) D. spicata vs. S. alterniflora, (C) J. roemerianus vs. S. alterniflora, (D) D. spicata vs. S. patens, (E) J. roemerianus vs. S. patens, and (F) J. roemerianus vs. D. spicata in choice experiments. Each point represents mean time spent within each habitat type by six snails within each choice trial. Line represents the 1:1 line which indicates no preference between the two choices. Statistically significant preferences (p < 0.05) are indicated with *.

Table 2 Statistical results from matched pairs t-tests for species, S. alterniflora state of senescence, and structural control preference experiments.

P-values < 0.05, as well as the option of the pair for which a significant preference was demonstrated, are denoted in bold.

Comparison	df	T	p	
Experiment 1: Species preference	
S. alterniflora vs. S. patens	9	4.796	<0.001	
S. alterniflora vs. D. spicata	8	3.471	0.007	
S. alterniflora vs.J. roemerianus	9	0.068	0.948	
S. patens vs. D. spicata	9	2.102	0.065	
S. patensvs. J. roemerianus	9	0.825	0.431	
D. spicata vs. J. roemerianus	9	1.525	0.162	
Experiment 2: S. alterniflora state of senescence preference	
Green vs. Yellow	9	2.854	0.019	
Green vs. Brown	9	2.631	0.027	
Yellow vs. Brown	9	0.755	0.470	
Experiment 3: Structural control experiments	
S. alternifloravs. dowel	4	2.984	0.041	
S. patens vs. dowel	4	1.802	0.146	
D. spicata vs. dowel	4	0.444	0.680	
J. roemerianus vs. dowel	4	1.733	0.158	

Experiment 2: S. alterniflora state of senescence preference

When snails were given a choice between S. alterniflora stems at different stages of senescence, snails significantly preferred partially senesced or dead S. alterniflora stems over live stems (Fig. 3, Table 2). Snails were observed on yellow (partially senesced) and brown (dead) stems 7.0 and 3.2 times more frequently, on average, than on green (live) stems (Figs. 3A, 3B). There was no consistent or significant difference in the proportion of time spent by snails on yellow (partially senesced) (13.7 ± 16.1%) compared to brown (dead) (9.8 ± 9.0%) stems (Fig. 3C.)

Figure 3 Snail preferences for Spartina alterniflora at different stages of senescence.

Percent of time snails spent on (A) yellow (partially senesced) vs. green (live), (B) brown (dead) vs. green (live), or (C) brown (dead) vs. yellow (partially senesced) Spartina alterniflora stems in choice experiments. Each points represents mean time spent within each habitat type by six snails within each choice trial. Line represents the 1:1 line which indicates no preference between the two choices. Statistically significant preferences (p < 0.05) are indicated by an asterisk.

Experiment 3: structural control

When snails were given a choice between plant stems and a structural control (wooden dowels), snails exhibited a variable preference response depending on which plant species was offered. Snails were 60 times more likely to be found on S. alterniflora (13.8 ± 19.8%) than on structural controls (0.2 ± 0.5%) (Table 2). In contrast, snails did not show a consistent or significant preference for any of the other three plant species over the structural controls (Table 2).

Discussion

As one of the most common organisms in the salt marshes of North America and a key trophic link in these ecosystems, the marsh periwinkle snail L. irrorata plays a significant role in salt marsh nutrient and energy flow (Silliman & Bertness, 2002). Although L. irrorata are thought to be dietary specialists for the smooth cordgrass Spartina alterniflora and fungal growth on this plant (Silliman & Zieman, 2001), they are also found on several other plant species (Hughes, 2012, Faillon, Wittyngham & Johnson, 2020), and their preferences between host plants remain unclear. Here, we combined field observations of snail densities among four common salt marsh graminoids—S. alterniflora, S. patens, D. spicata, and J. roemerianus—on nine sampling dates over a 20-month period with experiments of habitat choice among the same four species, to explore links in snail behavior and distributions.

Across almost two years of field surveys in Bay Batiste along the Louisiana Gulf Coast, snails were highly abundant on all four plant species, with densities comparable or higher on all species than reported on S. alterniflora in other studies along Atlantic and Gulf Coasts (McFarlin et al., 2015; Rietl, Sorrentino & Roberts, 2018). Furthermore, high densities were consistent across seasons, suggesting the persistence of snails within stands. This reflects spatial persistence of snails documented in Florida (Hamilton, 1978) and Texas (Vaughn & Fisher, 1992). Several prior studies of L. irrorata distributions reported only high snail densities in S. alterniflora stands (e.g., Watson & Norton, 1985; Silliman & Zieman, 2001). However, our findings more closely reflect those of Hughes (2012), who found that L. irrorata densities were highest in mixed stands of S. alterniflora and J. roemerianus, and Faillon, Wittyngham & Johnson (2020), who found comparable snail densities in adjacent stands of S. alterniflora and S. cyrosuroides. High L. irrorata densities in plant stands composed of species other than S. alterniflora suggests a broader habitat niche for the snail than previously assumed, motivating experimental examination of habitat preferences across plant species. Furthermore, these high densities may amplify L. irrorata’s importance in salt marsh trophic networks.

In species preference experimentation, L. irrorata demonstrated a significant preference for S. alterniflora over S. patens and D. spicata, but no clear preferences between S. alterniflora and J. roemerianus, nor between any combinations of S. patens, D. spicata, and J. roemerianus. Snail preference for S. alterniflora over other species of plant was expected, as L. irrorata was most abundant in S. alterniflora patches in the field. There may be several mechanisms underlying this preference, however. One possibility, as suggested in prior studies, is that S. alterniflora is a known food source for L. irrorata. Yet L. irrorata also derives nutrition from epiphytic microalgae (which can grow on the stalks of many plant species), and benthic algae and detritus in the marsh soils (Alexander, 1979; Watson & Norton, 1985). Furthermore when food sources are plentiful, such as is often the case for L. irrorata, snail distributions may be determined by other criteria, such as predator avoidance (Zaret & Suffern, 1976; Loose & Dawidowicz, 1994). Callinectes sapidus, the primary predator of L. irrorata, is a threat from below commonly found in the marsh tidal zone, but cannot access snails higher in the marsh canopy (Hughes, 2012). Thickness and rigidity of stalks may therefore play a considerable role in habitat selection when snail densities are high; a flexible plant may bend or collapse under the weight of many snails. Of the four plant species studied here, S. alterniflora has the widest average thickness (Hester, Mendelssohn & McKee, 2001), yet J. roemerianus has the most rigid stems (Eleuterius, 1976). Such attributes may explain the lack of a significant difference in time spent on S. alterniflora and J. roemerianus. While there is no evidence that J. roemerianus tissue serves as a food source for L. irrorata, L. irrorata may persist in J. roemerianus stands by grazing on epiphytic microalgae, benthic algae, and detritus. Conversely, both S. patens and D. spicata have thin stalks that, on several occasions observed during choice experiments, collapsed under the weight of multiple snails. Refuge-seeking behavior on sturdy stalks of S. alterniflora and J. roemerianus, particularly for larger snails, also would explain empirical snail observations on the four plant species: snail densities were not only higher, but snail shell lengths were also longer, in stands of S. alterniflora and J. roemerianus. Along with stem thickness and rigidity, stem height may influence host selection and predator avoidance by snails (Hughes, 2012) with S. alterniflora and J. roemerianus typically growing taller than S. patens and D. spicata in these habitats. Plant stem height was standardized within and across species in these experiments, but the combination of stem thickness, rigidity, and height may influence snail refuge-seeking behaviors in the field.

Widespread removal of snail predators may dramatically impact snail habitat preferences and grazing behaviors if these preferences are motivated by predator avoidance. Callinectes sapidus populations on the eastern seaboard of the USA declined significantly in the 20th century up to present day (Abbe & Stagg, 1996; Cole, 1998; Kahn & Cole, 1998; Lipcius & Stockhausen, 2002; Lycett et al., 2020), and a number of physical and anthropogenic factors influence crab distributions (Jivoff et al., 2017). Changes in crab population structure likely affect snail distributions between marsh graminoid taxa and across vertical (ground-to-canopy) gradients. However, L. irrorata may still express predator avoidance behaviors even if no predators are present (Hughes, 2012), which suggests that L. irrorata may select for rigid plant stalks even if predator abundance is low. Future work should be aimed at determining the relative roles of snail foraging and predation in governing snail occupancy patterns.

We explored L. irrorata preferences between live, partially senesced, and fully senesced S. alterniflora stalks to complement choice experiments of live-only plant stalks from four species. Here, snails spent more time on partially senesced and standing dead S. alterniflora than on live green S. alterniflora, but snails did not exhibit a preference between partially senesced and standing dead plant tissue. These findings are consistent with our predictions and prior evidence on snail litter preferences (Bärlocher & Newell, 1994). Experimental habitat preference is thought to be a good indicator of associated grazing behaviors in the field (Leighton, 1966; Keesey, Knaden & Hansson, 2015), and in 23 out of 65 trials involving any form of S. alterniflora tissue there was evidence of grazing (long radulations on tissue) after just twelve hours of trial (Fig. S2). While snails in these experiments may have selected host plants due to factors beyond grazing quality, our experimental results of snail habitat preferences here are consistent with our predictions, and prior work that showed L. irrorata graze more upon fully senesced S. alterniflora tissue than live S. alterniflora tissue (Bärlocher & Newell, 1994). Previous analyses of snail stomach materials found that less than 2% (Silliman & Zieman, 2001) and 3% (Alexander, 1979) of snail gut content was live green plant material. Grazing experiments conducted by Bärlocher & Newell (1994) suggested a preference for standing dead leaves, in both recently collected and powdered form, over respective forms of “yellow-green” leaves (defined as 25–30% green tissue remaining). Senescing S. alterniflora tissue has higher lipid content and concentrations of desired fungal epiphytes than green tissue, and soft, decaying tissue is easily digested compared to live tissue (Bärlocher & Newell, 1994; Silliman & Zieman, 2001).

Synthesizing results from our field studies and choice experiments in the context of prior work, it remains possible that bottom-up (food availability and quality), top-down (predator avoidance), or both sets of factors may drive snail decision-making and distributions. We speculate that snails may therefore face a series of hierarchical decisions in selecting a plant host: if food availability is low, preference may be exhibited for S. alterniflora, especially decaying or dead tissue. When food availability is high, snails may seek out tall or rigid stems (e.g., J. roemerianus) that may best serve as refuge from aquatic predators (Hughes, 2012). High snail densities on all four plant species, combined with selection for J. roemerianus at comparable rates as S. alterniflora, also lends evidence to a broader habitat niche for the snail than previously suspected. Acting as a habitat generalist, rather than interacting only with a single plant species, may indicate wider recruitment across the marsh platform. Given that adult L. irrorata do not disperse far beyond where they have passively settled in their planktonic larval form (Hamilton, 1978; Vaughn & Fisher, 1992), tolerance of multiple plant hosts may enable snail colonization of mixed-vegetation habitats. Furthermore, a broader habitat niche may also offer snails greater resiliency in the face of disturbance, such as salinity changes or climate change-induced sea level rise, which can alter the composition of plant communities (Morris et al., 2002). Persistence of snails in the face of disturbance is important given their central connectedness to the rest of the marsh food web. Yet given the strong preferences of the snail for decaying S. alterniflora tissue, presumably for high forage quality, broader habitat preferences may also create an ecological trap. If snails select for, and remain in, monotypic stands of J. roemerianus to avoid predation, they may experience lower food availability or nutritional content, as their grazing is limited to microalgae and detritus rather than plant tissue. Snail decision-making in light of multiple possible hosts may therefore both provide plasticity and vulnerability, depending on the heterogeneity of available plant hosts.

Conclusions

Animal behavior is an important process that influences habitat preferences and can determine the distribution of organisms in space and time. Here, we quantified natural densities of the marsh periwinkle L. irrorata on four common marsh graminoids, and explored the mechanisms driving such observations via habitat choice experiments. Our findings suggest that the habitat niche of L. irrorata may be broader than single-species specialization for S. alterniflora. Broad habitat tolerances may provide the snail greater resilience to the stressful, dynamic conditions of salt marshes, if both food sources and adequate predator refugia are available. Given the high abundance of this marsh omnivore, its behaviors likely have important implications for salt marsh nutrient and energy flow. With a habitat preference for decaying plants over live tissue, L. irrorata may control plant productivity only during peak growth when senesced tissue abundance is low. We suggest a key modulating role of snail behavior for the salt marsh trophic network, drawing upon our combination of empirical observations with preference experiments.

Supplemental Information

Supplemental Information 1 Aquaria design for habitat preference analysis, displayed during simulated daytime (left) and nighttime (right) conditions

Sets of S. alterniflora, S. patens, D. spicata, J. roemerianus and wooden dowels were mounted on opposite sides using Styrofoam inserts, and six snails were placed in the center of each arena. Cameras were mounted above to record snail movement.

Supplemental Information 2 Evidence of grazing upon S. alterniflora

Radulations were concentrated on leaves of S. alterniflora plants, and were observed on many of the plant segments used in experimentation.

Supplemental Information 3 Recorded snail behavior

Example recording of snails exploring arena before demonstrating a preference for Spartina alterniflora over Juncus roemerianus.

Supplemental Information 4 Field Data

Rows indicate data from field sampling collections of snails. Each row provides information on each snail collected in 0.25m x 0.25m single-species plots (n = 3 –5) from a well-studied marsh site along the northwestern shore of Bay Batiste, LA (29.4759° N, 89.8543° W) between May 2016 and January 2018. Collection Date is the day of the sampling event. Plant species is the single species of marsh vegetation (SA = Spartina alterniflora, SP = Spartina patens, DS = Distichlis spicata, JR = Juncus roemerianus) where the plot (0.25m x 0.25m quadrat) was placed. Plot indicates the quadrat identification for each sampling quadrat within a species (note: location “B” for SA was sampled in more than one plot on some occasions). Length is the measured shell length (in mm) for each snail collected (note 1: if no snails were collected within a plot, no rows will be listed for that plot; note 2: shell lengths were not measured in May 2016 which is indicated by “N/A” in the row). The density within a plot can be determined by totaling the number of rows with the same collection date/plant species/plot designation.

Supplemental Information 5 Experimental Data

Rows indicate data from experimental trials testing preference patterns for snails. Replicate-the replicate number for each unique treatment. Treatment-comparison of plant material (G=green, Y=yellow, B=brown) or species (Sa=Spartina alterniflora, Sp=Spartina patens, Ds=Distichlis spicata, Jr=Juncus roemerianus). Preference-the area of interest in the trial (on, Near, No pref) for each potential choice (A being the first listed choice and B being the second listed choice). Day or Night First –start time of trial observations (day or night). Daytime or Nighttime Period –indicates whether the given row corresponds to observations during the simulated daytime (6 hours), or during the simulated nighttime (6 hours), portion of the trial. Proportion Time –the proportion of time occupied in the Treatment and Preference area of interest indicated in the other columns of the row.

We thank the members of the Roberts lab, especially Ron Scheuermann, Madelyn Sorrentino, Nicole Farley, Jacqueline Levy, Ariella Chelsky, and Anthony Rietl, for assistance in preliminary field lab work, and the LUMCON community for a productive research atmosphere during an REU experience. We thank the Department of Biological Sciences at Dartmouth College, especially Drs. Matt Ayres, Celia Chen, and Hannah ter Hofstede for guidance and constructive criticism.

Additional Information and Declarations

Competing Interests

Author Contributions

Data Availability

The authors declare there are no competing interests.

David H. Klinges conceived and designed the experiments, performed the experiments, analyzed the data, prepared figures and/or tables, authored or reviewed drafts of the article, and approved the final draft.

Charles W. Martin conceived and designed the experiments, performed the experiments, analyzed the data, prepared figures and/or tables, authored or reviewed drafts of the article, and approved the final draft.

Brian J. Roberts conceived and designed the experiments, performed the experiments, analyzed the data, prepared figures and/or tables, authored or reviewed drafts of the article, and approved the final draft.

The following information was supplied regarding data availability:

The field data and experimental data are available in the Supplemental Files.

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
