# Peer review of "Ecological associations of the coastal marsh periwinkle snail Littoraria irrorata: field and laboratory evidence of vegetation habitat preferences"

_PeerJ, doi:10.7717/peerj.19071_

## Round 0.1 · original submission · Minor Revisions

Dear authors, please make the necessary changes. I hope that the reviewers will then approve the publication of this article as soon as possible.

Reviewer 1 ·

Basic reporting

The authors present a well-written manuscript with appropriate background provided. Figures and tables are appropriate. Data is shared.

Experimental design

The research question was clearly defined and the methods and analysis described were appropriate for the questions. Methods are described in sufficient detail to be repeatable.

I did have questions regarding the field data - were plant shoot density and shoot height recorded during collections? If yes, it might be useful to include.

I also had a concern regarding using wooden dowels in the structural control experiment. Many studies use plastic mimics rather than wood because plastic is inert and less chemically reactive, and less likely to pick up chemical cues. I don't see a problem with using wood dowels if they were only used once (i.e., all trials and treatment combinations were run at the same time). If the wood was used in repeated trials, some justification would be beneficial.

Validity of the findings

The data have been provided. The results are clear, and the conclusions are sound based on the results. The authors don't try to overinterpret the data.

Reviewer 2 ·

Basic reporting

Klinges et al. demonstrate that the marsh periwinkle uses at least four different plant species as habitats, though at various densities. Based on preference studies, snails show clear preference for S. alterniflora and dead tissue.

This is a good ecological study in the northern Gulf of Mexico on a dominant marsh snail. The research contributes to our understanding of the snail’s niche within the marsh. Below are comments to improve the manuscript.

The manuscript is clearly written, appropriately cited, and well-reasoned. See my comments below for suggested improvements.

Experimental design

Design is adequate, but see my comments below for questions to be addressed.

Validity of the findings

Findings are valid. Conclusions are clear and supported by the results.

Additional comments

Abstract

Round densities to whole numbers

Line 43: remove ‘highly’

Line 44: ‘nails’ should ‘snails’

Introduction

Lines 92-102: You could delete this paragraph and end the previous paragraph with a line like, “However, L. irrorata are associated with other saltmarsh plant species (citations).”

Line 106: ‘Should be Failon’

Methods

Lines 148-151: Were the plant species tested separately or together? For example, was a snail offered S. alterniflora vs. control in one container and S. patens vs. control in another or all 4 species with the control in one container?

Line 167: You can say plastic wrap if you want to avoid a brand name.

Line 177: The video is fantastic.

Line 179: What is the justification for counting a snail as choosing a habitat based on its location on the aquarium wall? What about on the floor of the aquarium? What about when a snail was in between the habitats? From the set-up, it appears that there is space for the snails to be in between. (Referencing the supplementary video would help here).

Discussion

An excellent discussion

---

## Round 0.2 · accepted · Accept

Dear authors, I congratulate you on the acceptance of this manuscript for publication.

Reviewer 1 ·

Basic reporting

Basic reporting is good.

Experimental design

Experimental design is good.

Validity of the findings

Findings are valid.

Additional comments

The authors made changes based on reviewer comments and addressed all reviewer concerns.